elLIFE

# An essential and NSF independent role for α-SNAP in store-operated calcium entry

Yong Miao[1], Cathrine Miner[1], Lei Zhang[1], Phyllis I Hanson[2], Adish Dani[1], Monika Vig[1]*

[1]Pathology and Immunology, Washington University School of Medicine, St Louis, United States; [2]Cell Biology and Physiology, Washington University School of Medicine, St Louis, United States

**Abstract** Store-operated calcium entry (SOCE) by calcium release activated calcium (CRAC) channels constitutes a primary route of calcium entry in most cells. Orai1 forms the pore subunit of CRAC channels and Stim1 is the endoplasmic reticulum (ER) resident $Ca^{2+}$ sensor. Upon store-depletion, Stim1 translocates to domains of ER adjacent to the plasma membrane where it interacts with and clusters Orai1 hexamers to form the CRAC channel complex. Molecular steps enabling activation of SOCE via CRAC channel clusters remain incompletely defined. Here we identify an essential role of α-SNAP in mediating functional coupling of Stim1 and Orai1 molecules to activate SOCE. This role for α-SNAP is direct and independent of its known activity in NSF dependent SNARE complex disassembly. Importantly, Stim1-Orai1 clustering still occurs in the absence of α-SNAP but its inability to support SOCE reveals that a previously unsuspected molecular re-arrangement within CRAC channel clusters is necessary for SOCE.

## Introduction

*For correspondence: mvig@
wustl.edu

Competing interests: The
authors declare that no
competing interests exist.

Reviewing editor: Vivek
Malhotra, Center for Genomic
Regulation, Spain

Store-operated calcium entry or SOCE is activated in response to the depletion of endoplasmic reticulum (ER) calcium stores, and constitutes a primary mechanism of calcium influx in excitable and non-excitable cells (*Parekh and Putney, 2005*). Store-operated calcium release activated calcium (CRAC) channels have been extensively studied for their role in the activation of nuclear factor of activated T cells (NFAT) regulated gene expression in lymphocytes and other cells (*Lewis, 2001*; *Crabtree and Olson, 2002*; *Feske et al., 2005*; *Parekh and Putney, 2005*; *Vig and Kinet, 2009*). Previously, genome-wide RNAi screens in *Drosophila* cell lines led to the identification of ER-resident STIMs as the store sensors and plasma membrane (PM) resident Orai/CRACMs as the pore forming subunit of CRAC channels (*Liou et al., 2005*; *Zhang et al., 2005*, *2006*; *Feske et al., 2006*; *Peinelt et al., 2006*; *Prakriya et al., 2006*; *Yeromin et al., 2006*; *Vig et al., 2006a*, *2006b*). Stim1, and its homolog Stim2, contain a calcium-sensing EF-hand like domain facing the ER lumen. Upon store-depletion, Stim1 oligomerizes and concentrates in regions of ER directly adjacent to the PM, frequently referred to as junctional ER (*Liou et al., 2005*; *Zhang et al., 2005*; *Stathopulos et al., 2006*; *Wu et al., 2006*; *Liou et al., 2007*; *Luik et al., 2008*; *Stathopulos et al., 2008*). While some factors involved in store-dependent and store-independent Stim1 clustering have been described (*Honnappa et al., 2009*; *Smyth et al., 2009*; *Srikanth et al., 2010*, *2012*; *Walsh et al., 2010b*; *Krapivinsky et al., 2011*; *Lewis, 2011*; *Singaravelu et al., 2011*), little is known about the final steps in CRAC channel activation. For instance, Orai1 multimers are thought to diffuse freely until trapped by Stim1 clusters in the ER–PM junctions, with active CRAC channels consisting of Orai1 hexamers (*Penna et al., 2008*; *Park et al., 2009*; *Madl et al., 2010*; *Walsh et al., 2010a*; *Hou et al., 2012*), however, it remains unexplored

**eLife digest** Calcium is an essential element for many biological functions. In particular, the movement of calcium ions through the cell membrane has a central role in many of the signalling pathways that cells use to communicate with other cells. Signals are produced by calcium ions both entering and leaving the cell, with information being contained in the rate, location, and duration of the flow of ions.

Calcium is stored inside cells in a structure called the endoplasmic reticulum, and when stores of calcium are low, special channels in the cell membrane called CRAC (calcium release activated calcium) channels are used to ferry more calcium ions into the cell. This process, known as store-operated calcium entry, relies on two important groups of proteins: the Stim proteins that sense when calcium stores are low; and, the Orai structural proteins that form the actual channel.

Previous work has shown that when the calcium stores are low, the Stim proteins—which reside in the endoplasmic reticulum—form clusters and these clusters then move to a part of the endoplasmic reticulum that is next to the cell membrane, where they join the Orai1 proteins to form larger clusters. However, to date it has been unclear whether Stim-Orai clustering at the cell membrane is sufficient for CRAC channels to open, or if additional steps are involved.

Miao et al. now show that another protein is involved in the formation of functional CRAC channels. Working with fruit fly cells, Miao et al. used genetic techniques to prevent the expression of various proteins that were thought to have a role in the movement of calcium ions through the cell membrane. One of these candidates, a protein called α-SNAP that is found in the internal fluid of the cell, was identified as having a central role in the import of calcium ions into the cell. Further work showed that α-SNAP re-organizes the Stim and Orai proteins to produce working CRAC channels.

whether mere trapping of Orai1 by Stim1 in ER–PM junctions is sufficient or additional molecular steps enable optimal activation of SOCE. We hypothesized that additional cytosolic proteins might be necessary to impart functionality to Stim1–Orai1 clusters that constitute the CRAC channel complex and adopted a candidate-based approach based on our earlier genome-wide RNAi screen (*Vig et al., 2006b*).

Here we identify a direct and unexpected role for α-SNAP in the activation of SOCE. α-SNAP is a cytosolic protein with tetra-tricopeptide repeat (TPR)-like helical domains that bridges N-ethylmaleimide sensitive fusion protein (NSF) to soluble NSF attachment protein receptor (SNARE) complexes to promote their disassembly and SNARE recycling (*Clary et al., 1990*; *Chang et al., 2012*). We find that α-SNAP directly and independently binds Stim1 and Orai1 to regulate the function and molecular composition of Stim1–Orai1 clusters that form the CRAC channel complex. This role for α-SNAP in SOCE is independent of its conventional role in facilitating NSF mediated SNARE complex disassembly. Thus, we have identified a novel role for α-SNAP and our data define a new α-SNAP dependent step in the physiological activation of SOCE via CRAC channels.

## Results

### α-SNAP depletion strongly reduces SOCE and NFAT activation

To test our hypothesis that additional proteins are needed to facilitate SOCE under physiological conditions, we used ~200 to 500 base pair long double stranded RNA (dsRNA) to deplete the expression of six genes that we previously identified as candidates in a genome-wide RNAi screen (*Vig et al., 2006b*) in *Drosophila* Kc cells. We found that knockdown of soluble NSF attachment protein (SNAP) strongly reduces SOCE by day 3 (*Figure 1A* and *Supplementary file 1*). Given the almost complete inhibition of SOCE in SNAP deficient *Drosophila* cells, we hypothesized that SNAP might engage in novel, SOCE specific protein–protein interactions that may or may not involve NSF and SNAREs. α- and β-SNAP are the two mammalian proteins most closely related to *Drosophila* SNAP (*Figure 1—figure supplement 1*). α-SNAP is ubiquitously expressed while β-SNAP expression is largely restricted to the brain. We depleted α-SNAP in HEK-293 (*Figure 1B*), Jurkat T cells (*Figure 1C*) and U2OS cells (*Figure 1—figure supplement 2*) using lentivirus-based RNAi constructs (*Supplementary file 1*) and

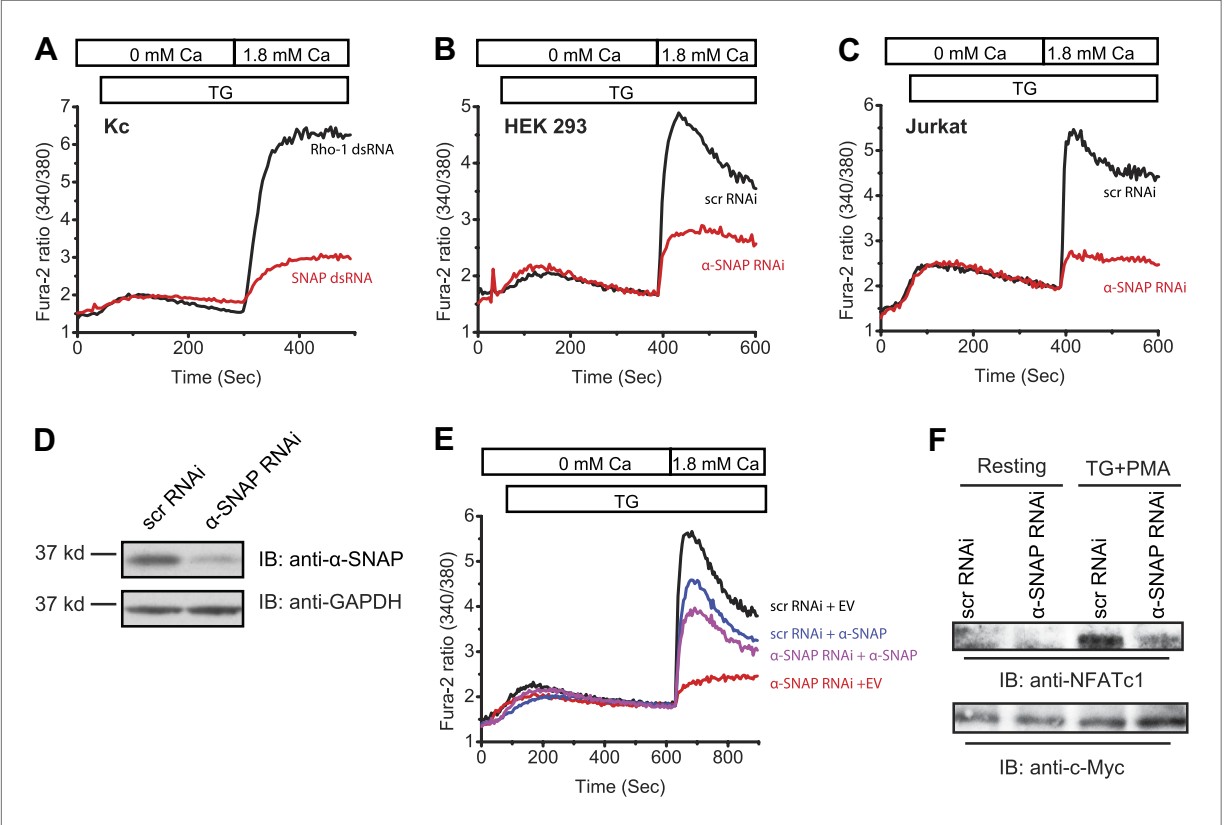

**Figure 1**. α-SNAP depletion inhibits SOCE and NFAT activation. (**A**) Average Fura-2 ratios of Drosophila Kc cells treated with dsRNA targeting *SNAP* (red) or *Rho-1* (black) for 3 days, and stimulated with 1 μM TG to measure SOCE using flexstation. (n > 3). (**B** and **C**) Average Fura-2 ratios of α-SNAP (red) or scramble (scr) (black) RNAi treated HEK 293 cells (**B**) or Jurkat T cells (**C**) stimulated with 1 μM TG to measure SOCE using flexstation. (n > 20) (**D**) A representative Western Blot for α-SNAP. WCLs of α-SNAP and scr RNAi treated HEK 293 cells were subjected to western blot analysis for each experiment using α-SNAP monoclonal antibody followed by anti-mouse secondary antibody. (**E**) Reconstitution of α-SNAP deficient cells with α-SNAP. Average Fura-2 ratios of RNAi treated HEK 293 cells stimulated with 1 μM TG to measure SOCE using flexstation, analyzed 3–4 days post RNAi transduction and 24 hr post-transfection with α-SNAP. (Black) cells transduced with scr RNAi and transfected with empty vector; (Red) cells transduced with α-SNAP RNAi and transfected with empty vector; (Blue) cells transduced with scr RNAi, transfected with α-SNAP; and (Purple) cells transduced with α-SNAP RNAi, transfected with α-SNAP. (n = 3) (**F**) Western blot for nuclear NFATc1 in RNAi treated Jurkat T cells. Nuclear extracts were prepared from RNAi treated Jurkat T cells, unstimulated, or stimulated with 1 μM TG and 10 ng/ml PMA and subjected to western blot using anti-NFATc1 antibody. c-Myc was used as a loading control. (n = 3)

The following figure supplements are available for figure 1:

**Figure supplement 1**. Alignment of *Drosophila* and human SNAP amino acid sequences.

**Figure supplement 2**. α-SNAP depletion using two different RNAi sequences in U2OS cells.

**Figure supplement 3**. Immuno-staining for α-SNAP in α-SNAP depleted and control cells.

**Figure supplement 4**. Reconstitution of α-SNAP deficient cells with β- or γ-SNAP.

**Figure supplement 5**. RNAi mediated depletion of γ-SNAP and measurement of SOCE.

found SOCE to be strongly inhibited by day three compared to cells treated with control RNAi. By day five, depletion of α-SNAP caused cell rounding in adherent cell lines; we therefore restricted our analysis to adherent cells at early time-points. We confirmed the efficiency of α-SNAP knockdown for each experiment on western blots of whole cell lysates (WCLs) (*Figure 1D*) and by immunostaining cells with anti-α-SNAP antibody (*Figure 1—figure supplement 3*). Importantly, when we reconstituted α-SNAP deficient HEK 293 cells with an RNAi resistant version of α-SNAP we found that SOCE was largely restored (*Figure 1E*). Defective SOCE in α-SNAP deficient HEK 293 cells could also be restored

by over-expressing β-SNAP (*Figure 1—figure supplement 4*). γ-SNAP is a third SNAP protein widely expressed in mammalian tissues but is less similar to *Drosophila* SNAP and α-SNAP than β-SNAP (*Figure 1—figure supplement 1*). γ-SNAP depletion failed to inhibit SOCE in HEK 293 cells (*Figure 1—figure supplement 5*) and γ-SNAP over-expression did not compensate for α-SNAP depletion (*Figure 1—figure supplement 4*).

SOCE is essential for the activation of nuclear factor of activated T cells (NFAT), which in turn regulates gene expression in lymphocytes and most other cells (*Lewis, 2001*; *Crabtree and Olson, 2002*). Western blot of nuclear extracts of α-SNAP deficient Jurkat T cells showed a strong reduction in nuclear translocation of endogenous NFATc1 in response to the store-depleting agent thapsigargin (TG) in combination with phorbol 12-myristate 13-acetate (PMA) (*Figure 1F*). These data demonstrate that α-SNAP is a novel regulator of SOCE and downstream signaling required for NFAT activation.

## α-SNAP binds Stim1 and Orai1 and co-localizes with CRAC channel clusters in ER–PM junctions

To understand how α-SNAP is involved in SOCE, we first asked whether α-SNAP binds Stim1 and/or Orai1. Co-immunoprecipitations (co-IPs) from lysates of store-depleted HEK 293 cells co-expressing either Stim1-Myc and YFP-α-SNAP (*Figure 2A*) or Flag-Orai1 and YFP-α-SNAP (*Figure 2B*) showed that α-SNAP co-precipitates with both Stim1 and Orai1 and vice versa. To determine if the interaction is direct, we incubated 100 nM purified α-SNAP with Orai1 and Stim1 immunoprecipitates (IPs) in vitro. α-SNAP bound to both Stim1 and Orai1 but not to an unrelated calcium channel, TRPC6 (*Figure 2C*).

We next sought to determine whether α-SNAP, a predominantly cytosolic protein, co-localizes with ER localized Stim1 and/or PM localized Orai1 under resting or store-depleted conditions. Some α-SNAP co-localized with Stim1 in almost all of the CFP-Stim1 expressing cells imaged even under resting conditions (*Figure 2D*). Upon store-depletion, we observed distinct co-localization of α-SNAP with Stim1 clusters in nearly 30–40% of cells (*Figure 2E*). Although there was no obvious co-localization between α-SNAP and Orai1-CFP expressed alone under resting (not shown) or store-depleted conditions (*Figure 2F*), α-SNAP co-localized at a higher frequency with clusters containing Stim1 as well as Orai1 in Stim1, Orai1 co-expressing stable cells, with nearly 75% cells showing distinct co-localization (*Figure 2G*). Taken together, these data show that a part of the total cellular pool of α-SNAP constitutively binds Stim1. Upon store-depletion, Stim1 likely brings α-SNAP to CRAC channel clusters where α-SNAP engages with Orai1.

## Regulation of SOCE by α-SNAP is NSF independent

α-SNAP is well known for its ability to promote SNARE complex disassembly and SNARE protein recycling by bridging NSF to SNARE complexes (*Clary et al., 1990*; *Hanson et al., 1995*; *Jahn et al., 1995*; *Xu et al., 1999*; *Marz et al., 2003*; *Barszczewski et al., 2008*; *Wickner and Schekman, 2008*; *Winter et al., 2009*; *Chang et al., 2012*). Therefore, α-SNAP depletion could affect Orai1 trafficking to the PM or Stim1 localization in the ER, thereby contributing to defective SOCE. Epifluorescence imaging of Orai1-YFP and YFP-Stim1 expressing α-SNAP depleted cells showed normal localization of both Orai1 and Stim1 (*Figure 3A and 3B*). To specifically quantify Orai1 protein levels at the PM, we tagged the extracellular loop of Orai1 with a α-bungarotoxin (BTX) binding site (BBS) (*Sekine-Aizawa and Huganir, 2004*) (*Figure 3—figure supplement 1*). Stable expression of Orai1-BBS-YFP followed by exogenous application of labeled BTX provided an estimate of cell surface Orai1 levels. α-SNAP depleted, resting, or store depleted cells did not show any significant difference in the levels of cell surface Orai1 when compared to controls (*Figure 3—figure supplement 2*). Consistently, immunostaining of α-SNAP depleted cells with an ER marker, calreticulin (*Figure 3—figure supplement 3*), and a Golgi-marker, giantin (*Figure 3—figure supplement 3*) showed no apparent changes in the ER-Golgi morphology suggesting that at the time points used in our studies, α-SNAP depletion had not perturbed overall organelle structure.

Since the only well defined function of α-SNAP, thus far, is to bridge NSF to SNARE complexes (*Clary et al., 1990*; *Chang et al., 2012*), we next asked whether NSF is involved in SOCE. We used inducible expression of a dominant negative NSF mutant, E329Q-NSF, which lacks the ATPase activity required for dissociating the SNARE complex (*Dalal et al., 2004*) and blocks transferrin flux through the endocytic pathway by 8 hr (*Figure 3C*). Surprisingly, SOCE was unaffected in E329Q-NSF expressing live and adherent cells even 20 hr after inducing the expression of the mutant (*Figure 3D and 3E*) and unlike α-SNAP, WT-NSF failed to co-localize with Stim1 clusters in ER–PM junctions (*Figure 3F*).

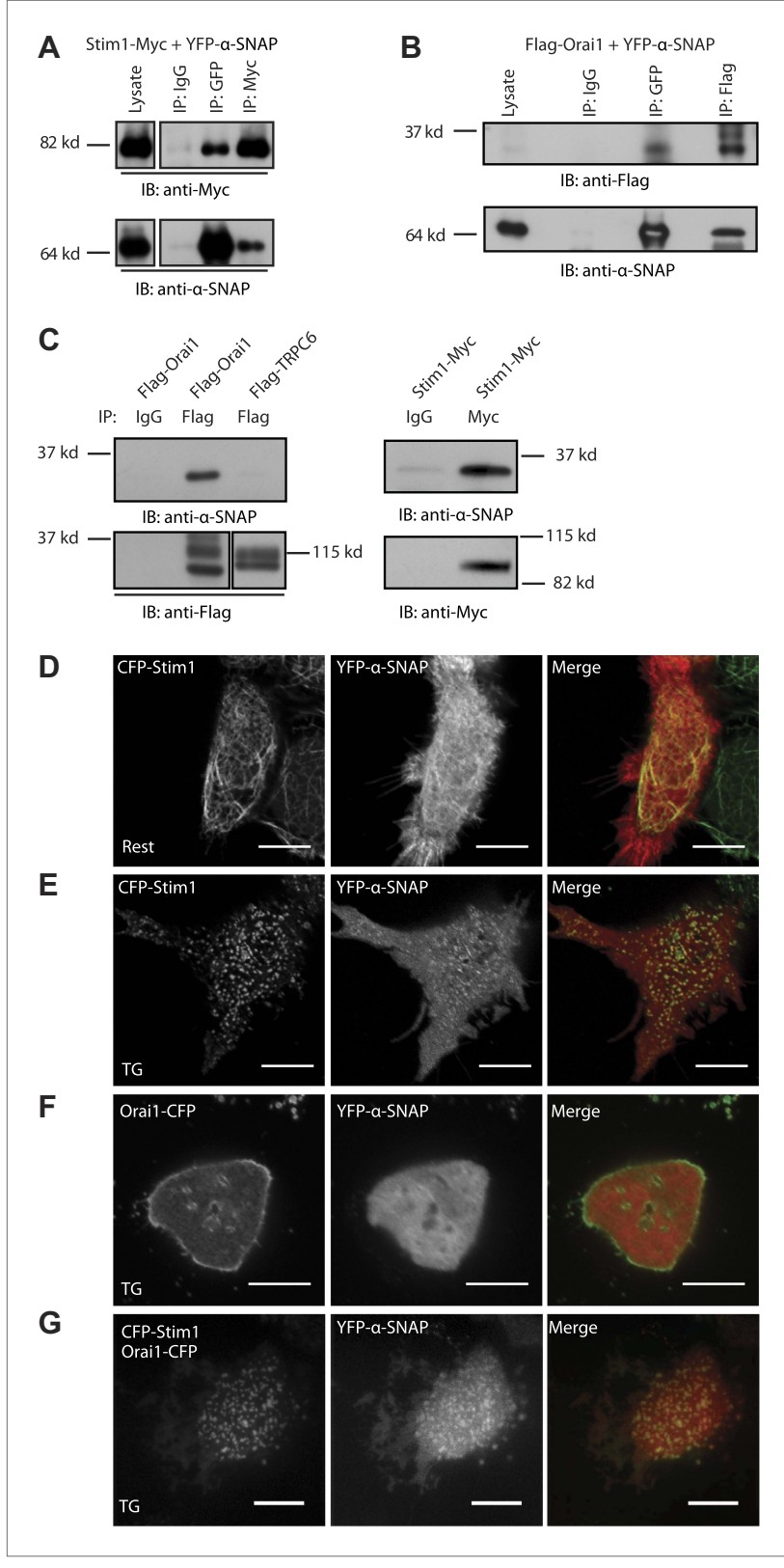

**Figure 2**. α-SNAP directly binds Stim1 and Orai1 and co-localizes with CRAC channel clusters.
(**A** and **B**) Co-immunoprecipitation of α-SNAP with Stim1 and Orai1. WCLs of store-depleted HEK 293 cells expressing Stim1-Myc and YFP-α-SNAP (**A**) or Flag-Orai1 and YFP-α-SNAP (**B**) were subjected to
*Figure 2. Continued on next page*

*Figure 2. Continued*

immunoprecipitation using anti-Myc, anti-Flag, or anti-GFP antibodies and Western Blot as indicated (n = 10). (C) In vitro binding of recombinant α-SNAP to full length Stim1 and Orai1. Flag-Orai1, Flag-TRPC6, and Stim1-Myc immunoprecipitates were incubated with purified α-SNAP protein (100 nM). Post-incubation, beads were washed, boiled, and subjected to western blot analysis using anti-α-SNAP antibody. (n > 10). (D, E and F) Representative confocal images of resting HEK 293 cells co-expressing YFP-α-SNAP and CFP-Stim1 (D) or store-depleted cells showing significant co-localization (E). Store-depleted HEK 293 cells co-expressing YFP-α-SNAP and Orai1-CFP (F). Scale bar 10 μm. (n = 3) (G) TIRF images of live, store-depleted HEK 293 cells, co-expressing CFP-Stim1, Orai1-CFP, and YFP-α-SNAP. Scale bar 10 μm. (n = 3)

Most importantly, reconstituting α-SNAP deficient cells with a previously reported mutant of α-SNAP, L294A (*Barnard et al., 1997*) (*Figure 4A*), that is unable to activate NSF and promote SNARE complex disassembly, failed to reconstitute transferrin flux through the endocytic pathway (*Figure 3G*) but fully restored SOCE (*Figure 3H*). Accordingly, like WT-α-SNAP, the L294A mutant co-immunoprecipitates Stim1 (*Figure 3I*) as well as Orai (*Figure 3J*). This clearly separates the function of α-SNAP in SOCE from its function in membrane trafficking demonstrating that α-SNAP regulates SOCE by a novel mechanism that is independent of its role in bridging NSF to SNARE complexes and more sensitive to its depletion.

## α-SNAP requires its hydrophobic loop for regulating SOCE

To explore the mechanism of regulation of SOCE by α-SNAP, we next set out to identify the domains involved in binding to the CRAC channel cluster and regulating SOCE by generating deletion and point mutations in α-SNAP as shown in *Figure 4A*. The N-terminal half of α-SNAP (1–160 aa) or α-SNAP-NT is predicted to contain a hydrophobic loop along with three putative TPR domains and the C-terminal half (161–295 aa) or α-SNAP-CT contains the fourth putative TPR domain (*Rice and Brunger, 1999*). Previous domain analysis of α-SNAP in the context of SNARE complex disassembly has shown that α-SNAP-CT but not α-SNAP-NT is necessary and sufficient for binding and activating the NSF ATPase activity (*Barnard et al., 1997*; *Rodriguez et al., 2011*). In contrast, upon store-depletion, α-SNAP-NT but not α-SNAP-CT co-localized with Stim1 clusters in CFP-Stim1 expressing cells (*Figure 4B*). Accordingly, α-SNAP-NT was able to partially restore SOCE in α-SNAP depleted cells (*Figure 4C*) suggesting that a specific domain within the N-terminal half (1–160 aa) of α-SNAP is crucial for binding to Stim1 and recruitment to the CRAC channel cluster.

To further define the involved region within the N-terminal fragment, we introduced four point mutations (F27S, F28S, L31S, F32S) inside a putative hydrophobic loop within the N-terminal half of α-SNAP (*Figure 4A*). Remarkably, reconstitution of α-SNAP deficient cells with this α-SNAP hydrophobic loop mutant (α-SNAP-LOOP) was able to restore transferrin uptake and recycling (*Figure 4D*) but failed to rescue SOCE (*Figure 4E*). Accordingly, α-SNAP-LOOP showed significantly reduced binding to Stim1 (*Figure 4F*) and Orai1 (*Figure 4G*) and failed to co-localize with Stim1 clusters in store-depleted HEK 293 cells stably expressing CFP-Stim1 (*Figure 4H*). Taken together these data demonstrate that the domains of α-SNAP required for SOCE are different from those essential for NSF activation and SNARE complex disassembly, conclusively demonstrating a distinct role for α-SNAP in SOCE.

## α-SNAP binds the CRAC activation domain of Stim1 and the C-terminal tail of Orai1 across the ER–PM junctional space

We next sought to identify the domains of Stim1 and Orai1 that α-SNAP binds within the CRAC channel cluster. Given the predominant cytosolic localization of α-SNAP, we expressed four different fragments of Stim1 cytosolic tail tagged with GST (*Lewis, 2011*; *Soboloff et al., 2012*); the full length cytosolic tail (CT), the inhibitory coiled coil domain CC1, the CRAC activation domain (known as CAD or SOAR), and the CAD plus CC1 domain in *Escherichia coli* and performed an in vitro binding assay with purified α-SNAP. α-SNAP showed strong binding to the CAD domain of Stim1 and faint binding to the CAD plus CC1 domain (*Figure 5A*). The CAD domain is a ~100 amino acid long, multifunctional cytosolic domain of Stim1, which when expressed with Orai1, engages with specific regions within the C- as well as the N-terminal cytosolic tails of Orai1 to activate SOCE (*Muik et al., 2009*; *Park et al., 2009*; *Yuan et al., 2009*). To identify the domains of Orai1 that bind α-SNAP,

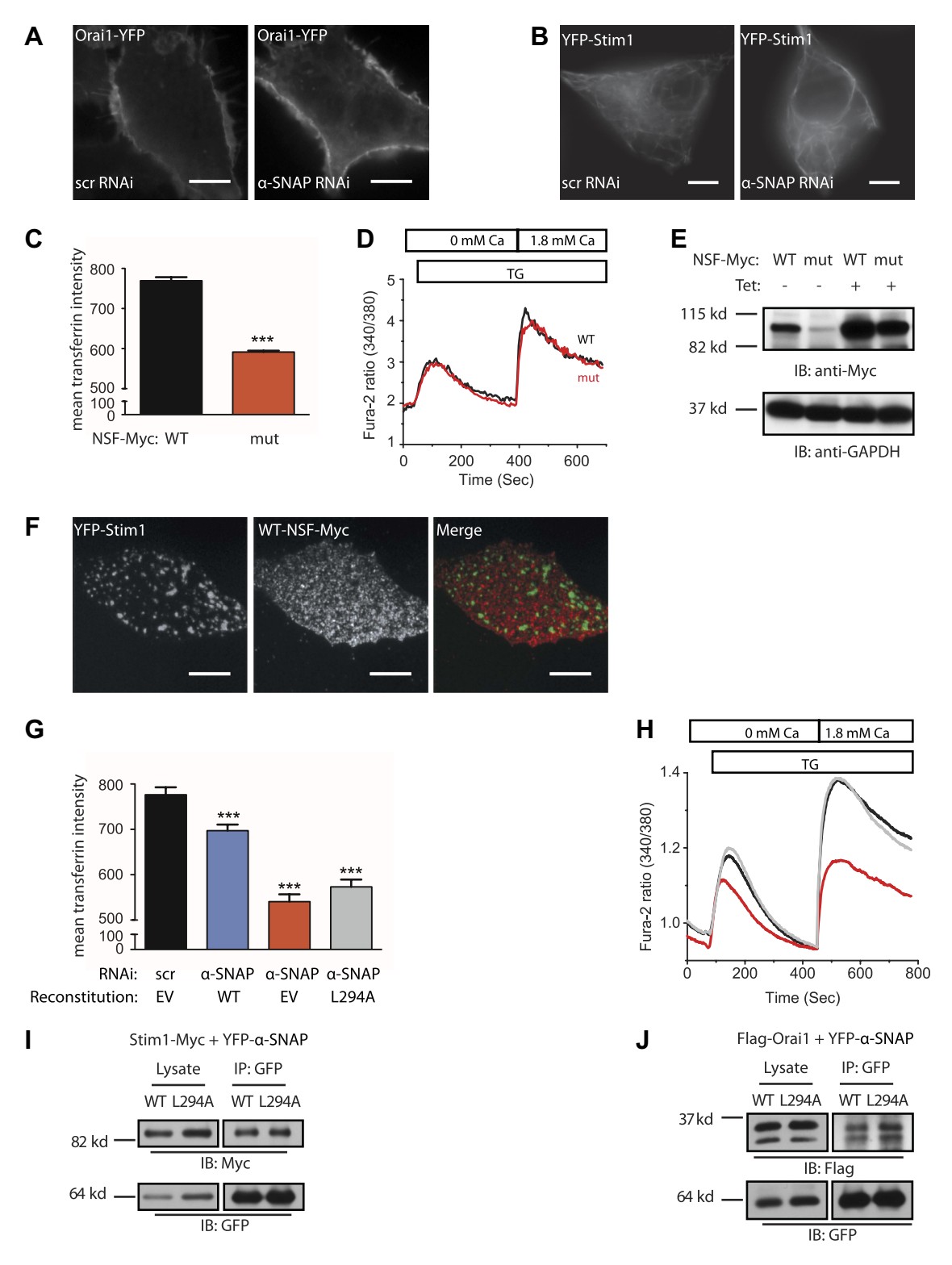

**Figure 3**. Regulation of SOCE by α-SNAP is NSF independent. (**A** and **B**) Resting localization of Orai1 and Stim1 in control and α-SNAP depleted cells. Epifluorescence images of α-SNAP and scr RNAi treated, resting HEK 293 cells stably expressing Orai1-YFP (**A**) or YFP-Stim1 (**B**). Scale bar 10 µm. (n > 5) (**C**) Quantification of intracellular transferrin-alexa 555 fluorescence. U2OS T-REx cells stably transfected with inducible WT-NSF (black) or E329Q-NSF (mut) (red) were induced with doxycycline for 8 hr. Post induction, cells were incubated with transferrin-alexa 555, washed, and fixed. Alexa 555
*Figure 3. Continued on next page*

*Figure 3. Continued*

fluorescence was quantified in cells using ImageJ. (n = 2 with ~80–100 cells scored per group) ***p<0.001 (**D**) SOCE in live and adherent U2OS cells expressing WT-NSF (black) vs E329Q-NSF (mut) (red). Fura-2 ratios averaged from cells induced with doxycycline for 16–20 hr and stimulated with TG. (n = 3) (**E**) Western blot on WCLs of WT- or E329Q-NSF expressing adherent U2OS cells analyzed in panel D. (**F**) Localization of WT-NSF with respect to Stim1 clusters in store-depleted cells. TIRF images of Myc-tagged WT-NSF expressing U2OS T-REx cells transiently transfected with YFP-Stim1 and stimulated with TG prior to imaging. Scale bar 10 µm. (**G**) Quantification of intracellular transferrin-alexa 555 fluorescence in RNAi treated HEK 293 cells. Cells transduced with scr RNAi and reconstituted with YFP alone (EV) (black bars) or transduced with α-SNAP RNAi and reconstituted either with YFP-alone (EV) (red), with YFP-tagged WT α-SNAP (blue) or with YFP-tagged α-SNAP-L294A mutant (gray). Post RNAi transduction and reconstitution, cells were incubated with transferrin-alexa 555, washed, and fixed. Alexa 555 fluorescence was quantified in YFP positive cells using ImageJ. (n = 2, with each experiment scoring ~100 cells) ***p<0.001 (**H**) Average single cell Fura-2 ratios of scr RNAi treated (black) or α-SNAP RNAi treated HEK 293 cells reconstituted with empty vector (red) or α-SNAP-L294A mutant (gray) showing SOCE in response to stimulation with TG. Cells were analyzed 3 days post RNAi transduction and 16 hr post-reconstitution. (n = 2) (**I** and **J**) Co-immunoprecipitation of Stim1 (**I**) and Orai1 (**J**) with WT α-SNAP or α-SNAP-L294A. WCLs of store-depleted HEK 293 cells co-expressing Stim1-Myc with YFP-α-SNAP or Stim1-Myc with YFP-α-SNAP- L294A (**I**) and Flag-Orai1 with YFP-α-SNAP or Flag-Orai1 with YFP-α-SNAP-L294A (**J**) were subjected to immunoprecipitation using anti-GFP antibodies and Western blot as indicated.

The following figure supplements are available for figure 3:

**Figure supplement 1**. Strategy for quantifying Orai1 levels in the plasma membrane.

**Figure supplement 2**. Quantification of cell surface Orai1 in α-SNAP depleted and control cells.

**Figure supplement 3**. Morphology of ER and Golgi in control and α-SNAP depleted cells.

we tagged the cytosolic N- and C-terminal tails of Orai1 with GST, expressed them in *E. coli* and found that purified α-SNAP predominantly binds to the C-terminal tail of Orai1 (Orai1-CT) although faint binding to the N-terminal tail of Orai1 (Orai1-NT) was also detected (*Figure 5B*).

Given that α-SNAP directly binds the two critical domains involved in the activation of SOCE via the CRAC channel complex, α-SNAP could facilitate a functional coupling between the ER localized Stim1 and Orai1 in the PM. To test this hypothesis and directly place α-SNAP in the molecular sequence of SOCE, we first asked whether the final molecular step of CAD domain mediated activation of SOCE via Orai1 is intact in α-SNAP depleted cells. To address this, we generated an inducible stable HEK 293 cell line, co-expressing Orai1 along with either full length Stim1 or just the soluble CAD domain of Stim1. As expected, α-SNAP depletion caused significant inhibition of SOCE in cells co-expressing full length Stim1 and Orai1 (*Figure 5C and 5D*). However, SOCE in cells co-expressing the cytosolic CAD fragment of Stim1 along with Orai1, although overall smaller, was resistant to α-SNAP depletion (*Figure 5E and 5F*). Similarly, basal calcium levels in cells co-expressing the cytosolic CAD fragment of Stim1 along with Orai1, although higher compared to WT HEK 293 cells, were unaffected by α-SNAP depletion (*Figure 5G*). Importantly, the lack of effect of α-SNAP depletion on CAD mediated constitutive and SOCE conclusively rules out any possible unknown effects on membrane potential or membrane trafficking of other components contributing to the activation of SOCE. Taken together, these data demonstrate that α-SNAP directly binds specific domains within Stim1 and Orai1 to enable a molecular step that precedes CAD mediated activation of SOCE in cells expressing full length Stim1.

## α-SNAP regulates the molecular composition of CRAC channel clusters in ER–PM junctions

Because α-SNAP binds the CAD domain of Stim1, we wondered whether α-SNAP contributes to the formation of store-depletion induced Stim1 clusters in the junctional ER. To observe these plasma membrane proximal regions of ER in live cells and to determine whether α-SNAP depletion affects Stim1 clustering in the junctional ER, we used Total Internal Reflection Fluorescence (TIRF) Microscopy. While the ability of Stim1 to oligomerize and translocate to the junctional ER upon store-depletion appeared normal in α-SNAP deficient cells, the size of Stim1-Orai1 clusters appeared larger (*Figure 6A*). We adopted a localized thresholding method to detect Stim1-Orai1 cluster boundaries (*Figure 6— figure supplement 1*), used it to quantify the size and intensity of individual Stim1–Orai1 clusters, and found that α-SNAP depletion increased the overall size of CRAC channel clusters (*Figure 6B*). Because Orai1 is thought to be largely dependent on Stim1 for its clustering in ER–PM junctions

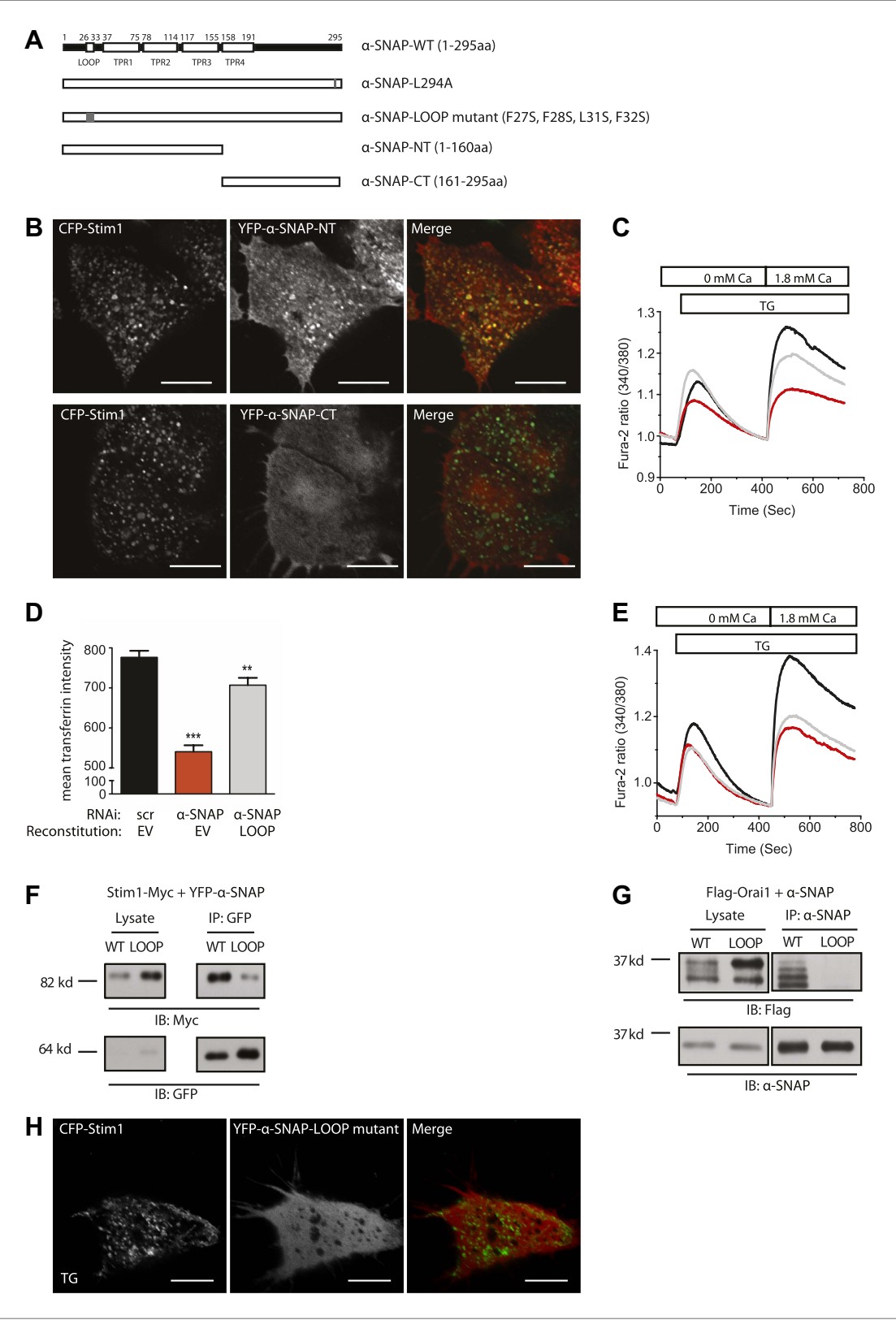

**Figure 4**. α-SNAP requires its hydrophobic loop for regulating SOCE. (**A**) Schematic showing domains of interest in human α-SNAP protein sequence and strategy for the generation of α-SNAP mutants. (**B**) Representative confocal images of store-depleted CFP-Stim1 expressing HEK 293 cells; transiently transfected with YFP-α-SNAP-NT (1–160 aa) or YFP-α-SNAP-CT (161–295 aa) Scale bar 10 µm (n = 3). (**C**) Average single cell Fura-2 ratios of scr

*Figure 4. Continued on next page*

*Figure 4. Continued*

(black) or α-SNAP RNAi treated HEK 293 cells, reconstituted either with empty vector (red) or with α-SNAP-NT (gray), showing SOCE in response to TG. Cells were analyzed three days post RNAi transduction and 16 hr post-reconstitution (n = 2). (**D**) Quantification of intracellular transferrin-alexa 555 fluorescence. HEK 293 cells were transduced with scr RNAi and reconstituted with YFP alone (black bars) or with α-SNAP RNAi and either reconstituted with YFP-alone (red) or with YFP-tagged α-SNAP-LOOP mutant (gray). Post RNAi transduction and reconstitution cells were incubated with transferrin-alexa 555, washed, and fixed. Intracellular alexa 555 fluorescence was quantified in YFP positive cells using ImageJ. (n = 2, with each experiment scoring ~100 cells) **p<0.01, ***p<0.001 (**E**) Average single cell Fura-2 ratios of scr (black) or α-SNAP RNAi treated HEK 293 cells reconstituted either with empty vector (red) or with α-SNAP LOOP mutant (gray), showing SOCE in response to TG. Cells were analyzed three days post RNAi transduction and 16 hr post-reconstitution. (n = 2) (**F** and **G**) Co-immunoprecipitation of Stim1 (**F**) and Orai1 (**G**) with WT-α-SNAP or α-SNAP-LOOP mutant. WCLs of store-depleted HEK 293 cells expressing Stim1-Myc and YFP-α-SNAP or YFP-α-SNAP-LOOP mutant (**F**) and HEK 293 cells expressing Flag-Orai1 and α-SNAP or α-SNAP-LOOP mutant (**G**) were subjected to immunoprecipitation using anti-GFP or anti-α-SNAP antibody and western blot as shown. (n = 2) (**H**) Representative confocal images of store-depleted CFP-Stim1 expressing HEK 293 cells; transiently transfected with YFP-α-SNAP-LOOP mutant. Scale bar 10 μm. (n = 3)

(*Liou et al., 2005*; *Zhang et al., 2005*; *Stathopulos et al., 2006*; *Wu et al., 2006*; *Liou et al., 2007*; *Varnai et al., 2007*; *Luik et al., 2008*; *Stathopulos et al., 2008*), we hypothesized that this increase in cluster size could result from an increase in the density of Stim1 in the junctional ER or could reflect a specific defect in the ability of Stim1 to efficiently co-cluster Orai1. To distinguish between these possibilities, we first quantified the average intensity of CFP-Stim1 in individual Stim1–Orai1 clusters. We did not find an appreciable difference in the average intensity of CFP-Stim1 estimated from Stim1–Orai1 clusters in control or α-SNAP depleted, HEK 293 cells stably expressing CFP-Stim1 and Orai1-YFP (*Figure 6—figure supplement 2*). These data suggest that α-SNAP depletion does not affect the density of Stim1 in junctional clusters. Surprisingly, we found a significant increase in the average Orai1-YFP intensity in α-SNAP depleted clusters (*Figure 6—figure supplement 2*), resulting in a significant decrease in the ratio of CFP (Stim1) to YFP (Orai1) across all CRAC channel clusters (*Figure 6C*) as well as in the majority of individual clusters (*Figure 6D*). These changes and their inhibitory effect on SOCE (*Figure 6—figure supplement 3*) suggest that α-SNAP regulates an active molecular rearrangement within CRAC channel clusters that is necessary for obtaining optimal Stim1/Orai1 ratios required for the physiological activation of SOCE through CRAC channels. Indeed, two independent recent studies that experimentally manipulated Stim1:Orai1 ratios in junctional clusters showed that the amplitude of calcium selective CRAC currents correlates well with the ratio of Stim1 to Orai1 or CAD to Orai1 within CRAC channel complex (*Mcnally et al., 2012*; *Hoover and Lewis, 2011*).

Given this reduction in the CFP (Stim1) to YFP (Orai1) ratio of α-SNAP deficient junctional clusters, we next wondered whether over-expression of α-SNAP would increase the Stim1/Orai1 ratio in clusters and thereby augment SOCE. Remarkably, over-expression of α-SNAP enhanced SOCE beyond what is typically seen in cells stably co-expressing Stim1 and Orai1 (*Figure 6—figure supplement 4*). Next, we examined the relative average intensities of CFP and YFP in junctional clusters of cells stably expressing CFP-Stim1, Orai1-YFP and transiently transfected with α-SNAP or empty vector. Concomitant with an increase in SOCE, over-expression of α-SNAP significantly enhanced the ratio of CFP (Stim1) to YFP (Orai1) across all junctional clusters (*Figure 6E*) as well as majority of individual clusters (*Figure 6F*). Notably, over-expression of α-SNAP did not affect total surface Orai1 levels in resting or store-depleted cells (*Figure 6—figure supplement 5*) or average CFP-Stim1 intensity per cell (data not shown).

In summary we have shown that α-SNAP directly binds Stim1 and Orai1 and co-localizes with CRAC channel clusters in the junctional ER. Within CRAC channel clusters, α-SNAP enables a crucial and previously unknown molecular re-arrangement that results in optimal Stim1:Orai1 ratios necessary for the physiological activation of SOCE (*Figure 7*).

## Discussion

We have identified a novel and direct role for α-SNAP in SOCE. Regulation of SOCE by α-SNAP is completely independent of its well-known role in NSF mediated SNARE complex disassembly. We demonstrate that Stim1–Orai1 clustering at ER–PM junctions is insufficient to fully activate SOCE unless followed by a α-SNAP dependent active molecular re-arrangement within CRAC channel clusters. Furthermore, our study opens the possibility that SNAP proteins are involved in organizing other membrane associated macromolecular complexes.

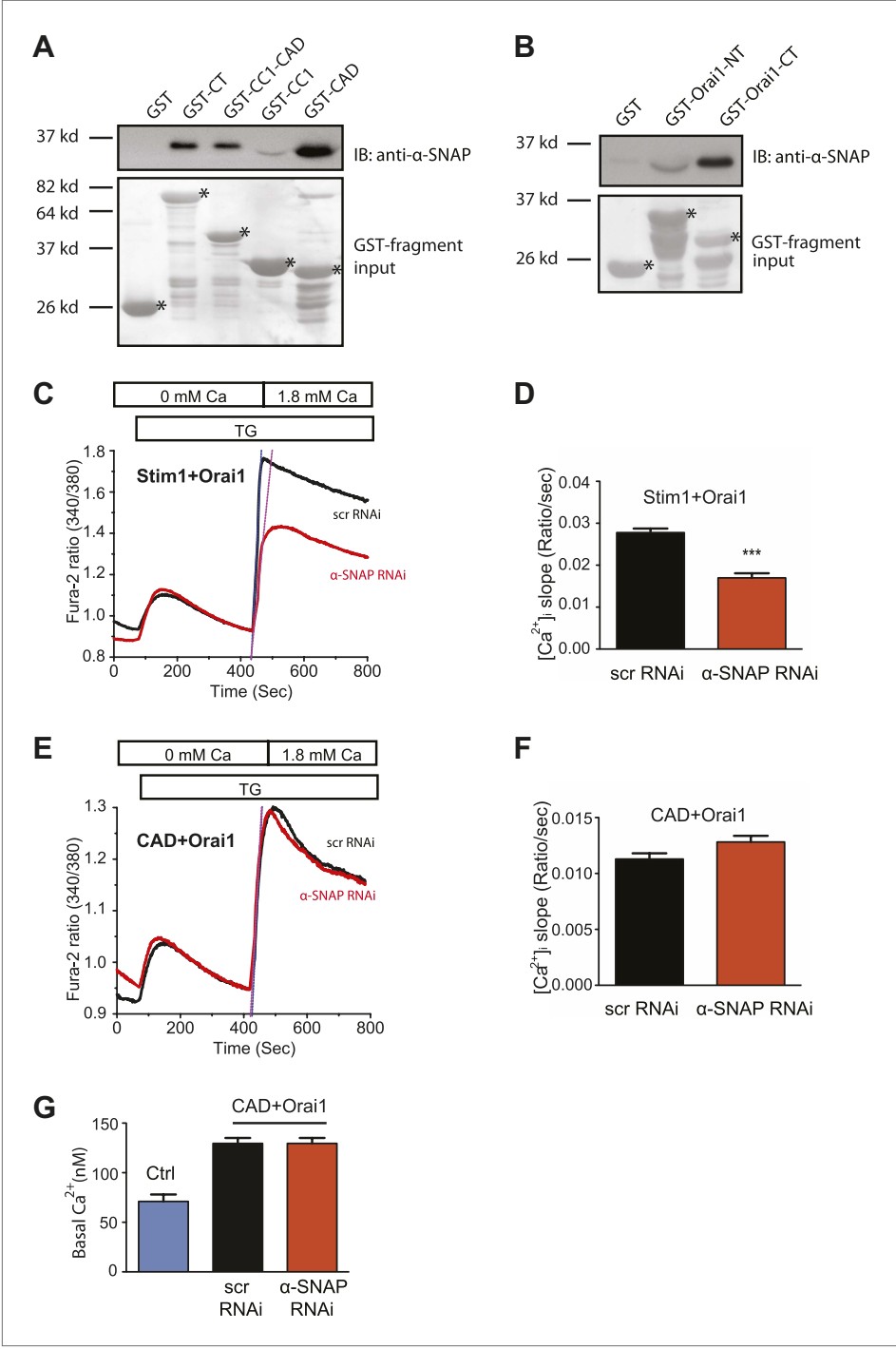

**Figure 5**. α-SNAP directly binds the CRAC activation domain of Stim1 and the C-terminal tail of Orai1. (**A**) In vitro binding of α-SNAP to cytosolic domains of Stim1. (Top panel) GST-tagged Stim1 fragments, expressed in *E.Coli* and immobilized on resin, were incubated with purified α-SNAP protein (10 nM). Post-incubation, beads were washed, boiled, and subjected to western blot analysis using anti-α-SNAP antibody. (Bottom panel) Ponceau S staining showing the input of GST-tagged fragments and their expected sizes. (n = 3) (**B**) In vitro binding of α-SNAP to cytosolic tails of Orai1. (Top panel) GST-tagged Orai1 cytosolic tails, expressed in *E.Coli* and immobilized on resin, were incubated with purified 10 nM α-SNAP protein as described above and subjected to western blot analysis using anti-α-SNAP antibody. (Bottom panel) Ponceau S staining showing GST-tagged fragment input and their expected sizes. (n = 3) (**C** and **E**) SOCE in RNAi treated HEK 293 cells stably expressing Orai1 and STIM1 (**C**) or Orai1 and CAD (**E**). Average, single cell Fura-2 ratios of cells transduced with α-SNAP (red) or scr (black) RNAi and
*Figure 5. Continued on next page*

*Figure 5. Continued*

stimulated with TG. (n = 3) (**D** and **F**) Measurement of the rate of increase in intracellular Ca²⁺. The rate of store-operated Ca²⁺ influx in (**C**) and (**E**) was estimated by measuring the maximal rate of initial rise in Fura-2 ratios after replenishing Ca²⁺ in the extracellular buffer. (p value***<0.001) (**G**) Basal intracellular Ca²⁺ concentration in unstimulated HEK 293 cells (Ctrl) or RNAi treated HEK 293 cells stably expressing Orai1 and CAD.

Several pieces of data presented here demonstrate a novel mechanism for α-SNAP mediated regulation of CRAC channel activity. Upon store-depletion, α-SNAP but not its usual binding partner, NSF, co-localizes with Stim1–Orai1 clusters. Secondly, dominant inhibition of NSF does not inhibit SOCE. α-SNAP depletion disrupts SOCE without affecting the resting and stimulated subcellular localization of Orai1 and Stim1. Furthermore, using deletion and point mutations in α-SNAP we show that the domains and residues crucial for facilitating SOCE do not overlap with those known to be essential for binding or activating NSF dependent SNARE complex disassembly (*Barnard et al., 1997*). Finally, the identification of an NSF independent role of α-SNAP is consistent with our previously reported primary genome-wide RNAi screen, where neither silencing of SNARE proteins nor NSF specifically inhibited SOCE in *Drosophila* S2 cells (*Vig et al., 2006b*).

A previous domain analysis of α-SNAP suggested that a hydrophobic loop within α-SNAP, although not directly essential for NSF activation, could function as a membrane attachment site to promote its recruitment to the SNARE complex (*Barnard et al., 1997*; *Winter et al., 2009*). Indeed, a similar role could explain the inability of the hydrophobic loop mutant of α-SNAP to efficiently bind Stim1 and Orai1 and co-localize with CRAC channel clusters to reconstitute SOCE.

Given that Stim1 and Orai1 can co-cluster in the junctional ER in α-SNAP depleted cells, our data suggest that Stim1 and Orai1 binding across ER–PM junctional space per se is not α-SNAP dependent. These data are consistent with many previous studies that have shown a direct interaction between Stim1 and Orai1 (*Yeromin et al., 2006*; *Vig et al., 2006a*) and identified the CAD domain of Stim1 as a critical domain involved in Stim-Orai1 clustering as well as SOCE activation (*Park et al., 2009*; *Yuan et al., 2009*). Interestingly, mutagenesis studies within the CAD domain have identified distinct residues that regulate Stim1 clustering vs activation of SOCE (*Park et al., 2009*; *Yuan et al., 2009*). Yet, the existence of non-functional Stim1–Orai1 coupling in a physiological context or the dependence of CRAC channel clusters on a third protein for regulating the final steps in SOCE activation has not been previously speculated. While several proteins have been previously identified to bind Stim1 (*Honnappa et al., 2009*; *Smyth et al., 2009*; *Walsh et al., 2010b*; *Krapivinsky et al., 2011*; *Lewis, 2011*; *Singaravelu et al., 2011*; *Soboloff et al., 2012*), only CRACR2A and Junctate have been shown to bind both Stim1 as well as Orai1 (*Srikanth et al., 2010*, *2012*). Of these, CRACR2A has been shown to bind CRAC channel clusters in a calcium dependent fashion and its depletion inhibits the formation of Stim1 clusters. Therefore, consistent with the current model of SOCE where Stim1–Orai1 clustering is considered necessary and sufficient for the physiological activation of SOCE, CRACR2A regulates the formation of Stim1–Orai1 clusters (*Srikanth et al., 2010*; *Lewis, 2011*; *Soboloff et al., 2012*).

We find that α-SNAP binds distinct and non-overlapping domains of Orai1 and Stim1 when compared to CRACR2A (*Srikanth et al., 2010*). More importantly, in contrast to CRACR2A, α-SNAP depletion inhibits SOCE without decreasing Stim1 clustering, the density of Stim1 in the junctional ER or the ability of Stim1 to bind Orai1. Therefore, for the first time, our studies have identified the existence of a crucial late step in the molecular sequence of SOCE that involves α-SNAP dependent active re-arrangement of Stim1 and Orai1 molecules within CRAC channel clusters necessary for the physiological activation of SOCE. We place the requirement for α-SNAP just after Stim1–Orai1 clustering but before CAD mediated activation of SOCE.

A fraction of α-SNAP binds Stim1 constitutively, however, binding to Orai1 likely stabilizes its interaction with and enhances its recruitment to the CRAC channel clusters since co-expression of the three proteins shows a significant increase in the percentage of cells showing co-localization of α-SNAP with CRAC channel clusters. In turn, co-expression of α-SNAP synergizes with Stim1 and Orai1 to amplify SOCE likely by ensuring that a higher percentage of Stim1–Orai1 clusters form functional CRAC channels. Interestingly, like in the case of α-SNAP depleted cells, we observed an increase in the size of CRAC channel clusters even in cells over-expressing α-SNAP (*Figure 6—figure supplement 6*), suggesting that the ratio of Stim1:Orai1 molecules within CRAC channel clusters determines the amplitude of SOCE rather than the size of clusters themselves or the absolute amount of Stim1 in the

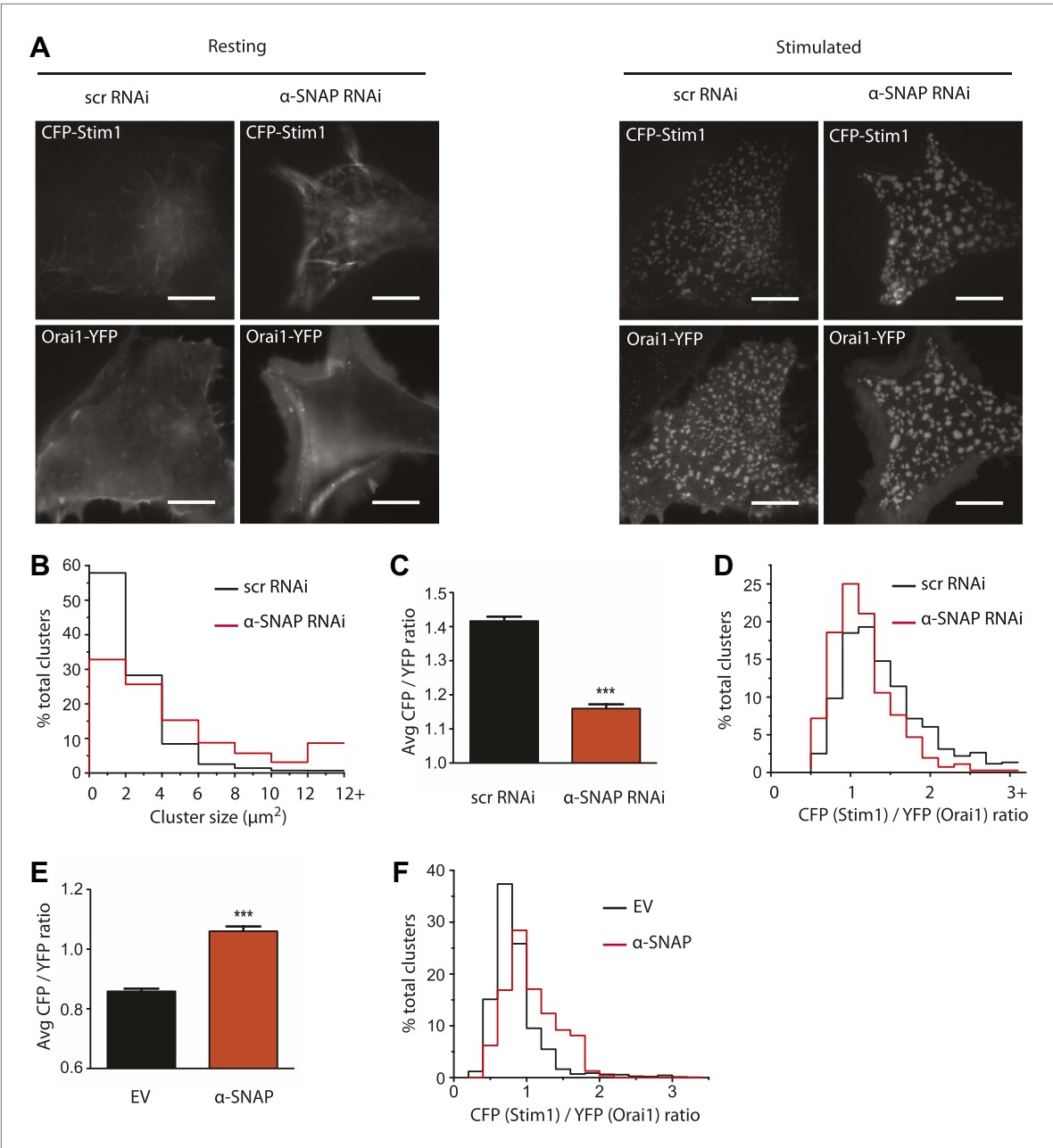

**Figure 6**. α-SNAP regulates the molecular composition of CRAC channel clusters in ER–PM junctions. (**A**) Stim1 translocation and Orai1 clustering in control and α-SNAP deficient cells. TIRF images of resting and store-depleted, control, and α-SNAP deficient HEK 293 cells stably expressing CFP-Stim1 and Orai1-YFP. Scale bar 10 μm. (n > 5) (**B**) Quantification of Stim1-Orai1 cluster size in control and α-SNAP deficient cells. TIRF images were acquired as in (**A**) and boundaries for individual Stim1-Orai1 clusters were detected (as shown in **Figure 6—figure supplement 1**), pooled from nine cells and compared between control (black) and α-SNAP depleted cells (red). Histograms show the size comparison of ~1000–1600 clusters. (**C** and **D**) Quantification of the relative intensities of CFP-Stim1 vs Orai1-YFP in CRAC channel clusters of RNAi treated cells. Stim1-Orai1 cluster boundaries were detected as described in (**B**) and average CFP vs YFP intensity of each cluster was calculated, pooled from nine cells and compared between control (black) and α-SNAP depleted cells (red). (**C**) Average CFP/YFP ratio of pooled clusters from RNAi treated cells and (**D**) CFP/YFP ratio of individual clusters from **Figure 6C**. (**E** and **F**) Independent quantification of relative intensities of CFP-Stim1 and Orai1-YFP in CRAC channel clusters over-expressing α-SNAP or empty vector. TIRF images were acquired from Orai1-Stim1 co-expressing HEK 293 cells transiently transfected with α-SNAP (red) or empty vector (black). Stim1-Orai1 cluster boundaries were detected and average intensity of CFP-Stim1 vs Orai1-YFP per cluster was calculated as described in **Figure 6B–D** above, pooled from six cells per group and compared. (**E**) Average CFP/YFP ratio of pooled clusters from α-SNAP (red) and empty vector (black) over-expressing cells. (**F**) CFP/YFP ratio of individual clusters from **Figure 6E**. (***p<0.001)

*Figure 6. Continued on next page*

*Figure 6. Continued*

The following figure supplements are available for figure 6:

**Figure supplement 1**. A representative Orai1-YFP image and its corresponding mask.

**Figure supplement 2**. Quantification of CFP-Stim1 and Orai1-YFP intensities in Stim1-Orai1 clusters of RNAi treated cells.

**Figure supplement 3**. α-SNAP depletion inhibits SOCE in Stim1-Orai1 over-expressing cells.

**Figure supplement 4**. α-SNAP co-expression augments SOCE in Stim1-Orai1 over-expressing cells.

**Figure supplement 5**. Quantification of cell surface Orai1 in α-SNAP over-expressing cells.

**Figure supplement 6**. Quantification of Stim1-Orai1 cluster size in α-SNAP over-expressing cells.

junctional ER. The increase in size could result from a disrupted stoichiometry between α-SNAP and a fraction of Stim1–Orai1 clusters in α-SNAP depleted as well as over-expressing cells. Alternatively, expansion in cluster size could result from an increase in the total amount of Stim1 in the junctional ER (*Wu et al., 2006*). Future studies employing sub-diffraction, single molecule imaging approaches would help elucidate the dynamics of α-SNAP dependent re-arrangement of Stim1-Orai1 molecules within junctional CRAC channel clusters.

## Materials and methods

### Plasmids and transfections

Orai1-Myc and Flag-Orai1, and Stim1-Myc plasmids have been described previously (*Peinelt et al., 2006*; *Vig et al., 2006b*). To generate Orai1-CFP and Orai1-YFP, human Orai1 was sub cloned into

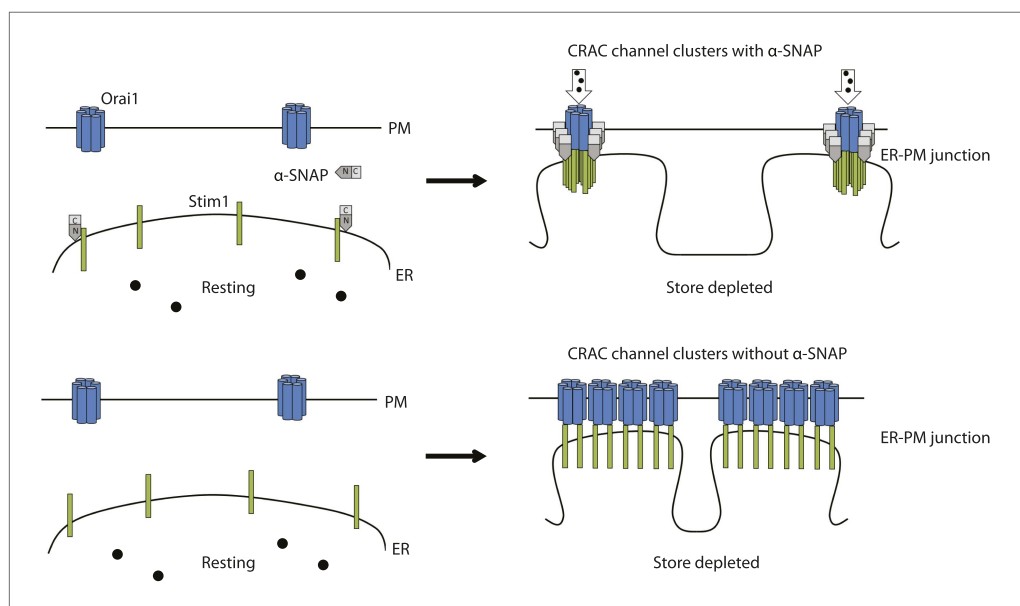

**Figure 7**. A hypothetical model of α-SNAP dependent re-arrangement of Stim1-Orai1 molecules within CRAC channel clusters in ER–PM junctions. Calcium ions (black circles), PM localized Orai1 hexamers (light blue) ER localized Stim1 (green), α-SNAP (grey). (Top left panel) Resting localization of Orai1 hexamers and Stim1 in α-SNAP sufficient cells. (Top right panel) α-SNAP sufficient, store-depleted cells form functional CRAC channel clusters with relatively high Stim1/Orai1 ratio. (Bottom left panel) Resting localization of Orai1 hexamers and Stim1 in α-SNAP deficient cells is unaffected. (Bottom right panel) α-SNAP deficient, store-depleted cells form bigger non-functional Stim1–Orai1 clusters with low Stim1/Orai ratio.

YFP-N1 and CFP-N1 (Clontech, Mountain View, CA). YFP-STIM1 and CFP-STIM1 were a gift from Dr Tobias Meyer's lab. Full-length human α-SNAP, β-SNAP, and γ-SNAP were amplified from human cDNA libraries and cloned into pcDNA/4TO/Myc-His (Invitrogen, Grand Island, NY) and pEYFP-C1 (Clontech) vectors. Stim1 cytosolic domains and Orai1 N- and C- tails were cloned into pGEX-4T-2 vector (GE Healthcare, Pittsburgh, PA) and named as GST-CT (235–685aa), GST-CC1-CAD (235–448aa), GST-CC1 (235–344aa), and GST-CAD (342–448aa) for Stim1 domains and GST-Orai1-NT (1–87aa) and GST-Orai1-CT (228–301aa) for Orai1 domains. All plasmid DNA transfections were done with Lipofectamine 2000 (Invitrogen) or Amaxa nucleofection kit (Lonza, Basel, Switzerland) according to manufacturer's protocol.

## Cell lines

All lentiviral RNAi transduction experiments were performed in WT HEK 293, Jurkat, U2OS (ATCC, Manassas, VA), and HEK 293 T-REx (Invitrogen) cell lines cultured in high glucose DMEM (Hyclone, Logan, UT) or RPMI (Hyclone) with 10% fetal bovine serum (Hyclone), 1% penicillin/streptomycin, and GlutaMax (Gibco, Grand Island, NY) or stable cell lines generated using these parent lines. CFP-Stim1 and Orai1-YFP were transfected into HEK 293 cells either alone or together, and stable lines were generated by cell sorting and G418 selection. HEK 293 T-REx (Invitrogen) cells were used to make double stable lines expressing Orai1-YFP with Tetracycline inducible Stim1-myc or CAD-myc according to the manufacturer's protocol. Orai1-BBS-YFP and Orai1-YFP was stably transfected into U2OS cells to estimate cell surface Orai1. U2OS T-REx cells stably expressing inducible WT-NSF or E329Q-NSF mutant were as described (*Dalal et al., 2004*).

## Antibodies and reagents

Anti-Flag antibodies (F7425 and F3165) were from Sigma (St. Louis, MO). Anti-GFP (A6455) and anti-NFATc1 antibodies were from Invitrogen and Biolegend (San Diego, CA) respectively. Supernatants from 9E10 hybridoma cultures were used to detect c-Myc. Anti-α-SNAP (clone 77.2) was from SySy (Goettingen, Germany), anti-calreticulin (PA1-24,485) from Affinity BioReagents (ABR, Golden, CO), and anti-giantin (PRB-114C) from Covance (Princeton, NJ). Anti-mouse IgG AF488 or anti-rabbit IgG Cy3 secondary antibodies were purchased from Jackson Immunoresearch (West Grove, PA). Transferrin-alexa 555 was from Invitrogen. All other reagents were from Sigma Aldrich.

## Lentiviral transductions

shRNA cloned into pLKO.1-puro vectors were purchased from the RNAi Core at Washington University and co-transfected with psPAX packaging and VSV-G envelope plasmids into HEK 293-FT (Invitrogen) cells for generating supernatants containing infectious viral particles. Viral supernatants were collected at 48 hr, filtered and used to transduce target cells along with 4 µg/ml polybrene. Cells transduced with viral supernatant were selected using 2 µg/ml puromycin.

## Single Cell Ca²⁺ imaging

Cells plated on coverslips were loaded with 1 µM Fura-2 AM in Ringer's buffer (135 mM NaCl, 5 mM KCl, 1 mM $CaCl_2$, 1 mM $MgCl_2$, 5.6 mM Glucose, and 10 mM Hepes, pH 7.4) for 40 min in the dark, washed, and used for imaging. An Olympus IX-71 inverted microscope equipped with a Lamda-LS illuminator (Sutter Instrument, Novato, CA), Fura-2 (340/380) filter set (Chroma, Bellows Falls, VT), a 10X 0.3NA objective lens (Olympus, UPLFLN, Japan), and a Photometrics Coolsnap HQ2 CCD camera was used to capture images at a frequency of ~1 image pair every 2 s. Data were acquired and analyzed using MetaFluor (Molecular Devices, Sunnyvale, CA), Microsoft Excel, and Origin software. At least 40–50 cells were imaged per group in each experiment.

## Calcium imaging using flexstation III

Cells (~100,000/well) plated in 96-well plates were loaded with Fura-2 AM as described above. Fura-2 excitation ratios were measured by alternatively exciting the dye at 340 and 380 nm, at a frequency of ~1 image pair every 4 s and collecting emission at 510 nm using Flexstation III (Molecular Devices) equipped with SoftMax Pro 5 software (Molecular Devices). Cells were lysed in each well at the end of the run to compare the efficiency of dye loading across different groups. Data were analyzed using Softmax Pro 5, Microsoft Excel, and Origin software.

## Preparation of nuclear extracts

Resting and stimulated cells were washed in cold Ringer's buffer, resuspended in 100 µl of chilled hypotonic buffer (10 mM HEPES [pH 7.9], 10 mM KCL, 0.1 mM EDTA, 1 mM DTT, and protease inhibitors)

and swelled on ice for 15 min. Subsequently, 0.6% NP-40 was added and cells vortexed and centrifuged. The nuclear pellet was washed three times and resuspended in 50 µl cold protein extraction buffer (20 mM HEPES [pH 7.9], 0.4 M NaCl, 1 mM EDTA, 1 mM DTT, and protease inhibitors), vortexed, centrifuged, and extracts were subjected to Western Blot analysis.

## Co-immunoprecipitation, *E. coli* expression, in vitro binding and Western Blotting

HEK 293 cells were lysed using buffer containing 1% CHAPS or 1% NP-40, 150 mM NaCl, 50 mM Tris, pH 8.0, and protease inhibitors. Unless otherwise mentioned, 1/10th of the whole cell lysate (WCLs) were used in the lysate input lanes. The remaining WCLs were pre cleared for 1 hr and incubated with appropriate primary antibodies overnight at 4°C followed by Protein A Sepharose beads for 1 hr at 4°C. The immunoprecipitates were boiled and subjected to western blotting using appropriate primary and anti-mouse IgG or anti-rabbit IgG secondary antibodies.

For in vitro binding to full length proteins, purified recombinant α-SNAP was incubated with Flag- or myc-Orai-1 or Stim1-myc immunoprecipitated from HEK 293 cells and bound to Protein A Sepharose beads for 1 hr at 4°C. Post-incubation, beads were washed three times and protein complexes eluted by boiling in SDS containing sample buffer and subjected to SDS-PAGE and western blotting.

For in vitro binding to GST-fused Stim1 and Orai1 fragments, proteins expressed in Lemo21 *E. coli* (NEB C2528H, Ipswich, MA) were purified using glutathione sepharose and incubated with purified His-α-SNAP (10–100 nM) for 1 hr. After washing, the proteins were eluted by boiling beads with SDS-sample buffer and subjected to SDS-PAGE and western blotting.

## Immunostaining

Cells were plated on poly-Lysine coated coverslips (VWR, Radnor, PA and Fisher, Pittsburgh, PA) or 35-mm glass bottom culture dishes (MatTek, Ashland, MA), fixed using 2% PFA for 15 min, permeabilized and blocked in 0.1% saponin, 3% BSA in PBS for 45 min, and then stained with appropriate primary and secondary antibodies and imaged using a confocal or a wide field microscope.

## Transferrin uptake and recycling assay

Cells plated onto coverslips were washed three times with serum free DMEM. Transferrin-alexa 555 (Invitrogen) was diluted at 25 µg/ml in serum free DMEM and incubated with cells at 37°C for 30 min, followed by three quick washes with serum free DMEM, pH 5.5, to remove surface bound transferrin. Subsequently cells were fixed with 2% PFA and imaged using IX-71 microscope with a 60X 1.25 NA objective lens (Olympus, UPFLN). Cell boundaries were identified using YFP fluorescence and total alexa 555 fluorescence per cell was quantified using ImageJ software.

## Confocal imaging

Confocal images were captured on an inverted Olympus Fluoview FV1000 microscope using a 60X 1.4 NA objective. Thin optical sections of CFP and YFP were obtained by sequentially scanning 440 nm and 515 nm lasers respectively. Image analysis was performed using ImageJ (NIH) and Adobe Photoshop.

## Wide-field epifluorescence and TIRF microscopy

Wide-field epifluorescence images were captured using the Olympus IX-71 microscope described above with a 60X 1.25 NA objective lens (Olympus, UPFLN), FITC/YFP/Cy3 dichroics (Brightline, Semrock, Rochester, NY), and the Micro-Manager software. Image analysis was performed using ImageJ (NIH) and Adobe Photoshop.

TIRF images were captured on a Nikon Eclipse TiE microscope fitted with the Nikon perfect focus system (TiE-PFS) for focus stabilization, a motorized stage (Marzhauser), and using an objective type TIRF geometry. Illumination lasers, 405 nm (Coherent, Cube), and 488 nm (Coherent, Sapphire) were individually shuttered (Uniblitz, Vincent Associates), combined, expanded, collimated, and focused at the back focal plane of a 100X 1.4 NA objective (Olympus UPLSAPO). CFP or YFP images were sequentially acquired by the same objective and separated using a dichroic set ZT405/488/561/640rpc-ZET405/488/561/640m and filtered using ET450/40m or ET525/50m emission filters (Chroma). Images were captured using custom software on an EM-CCD camera (Andor, iXon DU-897). Orai1-YFP and CFP-STIM1 expressing cells were first imaged in the resting stage and their positions marked. After addition of 1 µM Thapsigargin (TG), the same cells were imaged to avoid any bias in selection of cells based on cluster brightness or morphology.

## Quantification of size and fluorescence intensity of clusters

TIRF images were uniformly scaled to generate a binary image mask (See *Figure 6—figure supplement 1*) and Orai-Stim cluster boundaries were detected by adapting Otsu's thresholding method to perform localized thresholding in Matlab (http://www.mathworks.com/help/images/ref/graythresh.html). The cluster mask was applied to corresponding Orai1-YFP and CFP-Stim1 images to obtain the size and average CFP or YFP intensity per cluster. Cluster size and intensity data were pooled from roughly nine cells per group and plotted as a histogram.

## Bungarotoxin binding assay

A 13 amino acid bungarotoxin binding site (BBS) WRYYESSLEPYPD (*Sekine-Aizawa and Huganir, 2004*) was inserted into Orai1 extracellular loop between G207 and Q208. Orai1-BBS-YFP was stably expressed in U2OS cells. After RNAi treatment or transient over-expression of α-SNAP, cells were labeled with Alexa-647 conjugated α-bungarotoxin (BTX) (Invitrogen) for 30 min on ice. Samples were analyzed using FACS Caliber and FlowJo software.

## Acknowledgements

We thank Andrey Shaw and Barry Sleckman for critical reading of this manuscript. The Genome Institute at Washington University provided lentiviral RNAi constructs.

## Additional information

### Funding

| Funder | Grant reference number | Author |
|---|---|---|
| Rheumatic Diseases Core Center Piot and Feasibility Award | NIH P30AR048335 | Monika Vig |

The funders had no role in study design, data collection and interpretation, or the decision to submit the work for publication.

### Author contributions

YM, CM, Performed experiments, generated reagents and analyzed the data; LZ, Performed analysis of Stim1-Orai1 clusters using a custom written software; PIH, Contributed WT and mutant NSF stable lines, purified α-SNAP and α-SNAP antibody and helped with drafting and revising the article; AD, Designed and helped with TIRF imaging, analysis of Stim1-Orai1 clusters, drafting and revising the article; MV, Performed experiments, generated reagents and analyzed the data, designed the study and drafted the article

## Additional files

### Supplementary files

• Supplementary file 1. dsRNA and shRNA sequences used in this study.

### Major dataset

The following previously published dataset was used:

| Author(s) | Year | Dataset title | Dataset ID and/or URL | Database, license, and accessibility information |
|---|---|---|---|---|
| Vig M, Peinelt C, Beck A, Koomoa DL, Rabah D, Koblan-Huberson M, Kraft S, Turner H, Fleig A, Penner R, Kinet JP | 2006 | CRACM1 Is a Plasma Membrane Protein Essential for Store-Operated Ca2+ Entry-List of Reported Hits by Monika Vig | http://www.flyrnai.org/cgi-bin/RNAi_public_screen.pl?project_id=64; https://pubchem.ncbi.nlm.nih.gov/assay/assay.cgi?aid=720505 | Publicly available at DRSC published screens (http://www.flyrnai.org/DRSC-PTO.html); screens for genes relevant to Signal Transduction |

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
