## [Decision Letter]

Thank you for sending your work entitled “An Essential and NSF Independent Role For α-SNAP In Store-operated Calcium Entry” for consideration at *eLife*. Your article has been favorably evaluated by a Senior editor and 2 reviewers, one of whom is a member of our Board of Reviewing Editors.

The Reviewing editor and the other reviewer discussed their comments before we reached this decision, and the Reviewing editor has assembled the following comments to help you prepare a revised submission.

Your work presents an investigation of the involvement of α-SNAP in the organization of two components of Store Activated Calcium Entry (SOCE): Orai1 and Stim. Orai1 and Stim1 interact and cluster at the ER-PM junction, and constitute the active CRAC channel. The ratio of Orai1:Stim1 is crucial for SOCE as reported recently (McNally et al., 2012). The surprising finding here is that α-SNAP function regulates the ratio of Orai1 and Stim1 clusters and, more importantly, this function is independent of NSF. The data on the requirement of α-SNAP in this process and its interaction with Orai1 and Stim1 represent a potentially important step in our understanding of SOCE. However, the data does not reveal how α-SNAP controls the size of the clusters containing Orai1 and Stim1. We would be willing to consider a revised version provided it contains new data that addresses the following two major issues.

1) Does the L294A α-SNAP mutant, which does support SOCE but not SNARE function, also bind STIM1 and Orai1 like WT?

Likewise, can the N-terminal half of α-SNAP, which colocalizes with STIM1 (Figure 4), rescue SOCE as well? This would also help to address whether the entire functional domain is contained in the N-terminal fragment of α-SNAP.

2) In Figure 5, you show that recombinant α-SNAP directly binds GST-fusion proteins containing Orai1 and Stim1 fragments. However, it is not clear which parts of α-SNAP are required for the interaction with Orai1 and Stim1. Thus, the α SNAP LOOP mutant might affect selectively binding to either Stim1 or Orai1, which is important to test.

Furthermore, based on Figure 5, it appears that the binding of α-SNAP to the CAD domain of Stim1 is stronger than interactions with the CT domains containing the CC1 fragment. Thus, the binding of α-SNAP to the CAD domain may relieve a potential inhibition caused by the CC1 domain. Interestingly, upon expression of the CAD domain itself, depletion of α-SNAP does not reduce SOCE. Thus, a simple working hypothesis would be that binding of α-SNAP to the CAD domain of Stim1 releases a potential inhibition by the CC1 domain. This can be tested by using recombinant GST–CT Stim1 (already employed in Figure 5) and add increasing amounts of recombinant Orai 1 to obtain a rough estimate of the binding affinity. Addition of increasing amounts of α-SNAP should then increase the affinity of the Stim1–Orai1 reaction or alter the stoichiometry. In a control reaction, α-SNAP could be added to a reaction containing GST–CAD instead of GST–CT Stim1. Assuming that the CAD domain is already active, a-SNAP would not affect the GST–CAD–Orai1 interaction. This experimental set up could provide a mechanistic model of how α-SNAP affects SOCE.

---

## [Author Response]

*1) Does the L294A α-SNAP mutant, which does support SOCE but not SNARE function, also bind STIM1 and Orai1 like WT*?

Similar to WT α-SNAP, the L294A α-SNAP mutant can co-IP Orai1 as well as Stim1. These data have now been added to Figure 3 panels I & J.

*Likewise, can the N-terminal half of α-SNAP, which colocalizes with STIM1 (Figure 4), rescue SOCE as well? This would also help to address whether the entire functional domain is contained in the N-terminal fragment of α-SNAP*.

As suggested, we tested the ability of the N-terminal fragment of α-SNAP to reconstitute SOCE. Remarkably, the N-terminal fragment of α-SNAP can partially restore SOCE in α-SNAP depleted cells. These data suggest that the N-terminal fragment of α-SNAP is likely to contain a major part of the SOCE regulatory domain. These data have been included in Figure 4 panel C.

*2) In Figure 5, you show that recombinant α-SNAP directly binds GST-fusion proteins containing Orai1 and Stim1 fragments. However, it is not clear which parts of α-SNAP are required for the interaction with Orai1 and Stim1. Thus, the α SNAP LOOP mutant might affect selectively binding to either Stim1 or Orai1, which is important to test*.

We tested the ability of α-SNAP LOOP mutant to co-IP Orai1 and found a significant defect when compared to WT α-SNAP. These data are consistent with the inability of the loop mutant to bind Stim1, lack of co-localization with CRAC channel clusters, and its inability to reconstitute SOCE in α-SNAP depleted cells. The data have been added to Figure 4 panel G.

*Furthermore, based on Figure 5, it appears that the binding of α-SNAP to the CAD domain of Stim1 is stronger than interactions with the CT domains containing the CC1 fragment. Thus, the binding of α-SNAP to the CAD domain may relieve a potential inhibition caused by the CC1 domain. Interestingly, upon expression of the CAD domain itself, depletion of α-SNAP does not reduce SOCE. Thus, a simple working hypothesis would be that binding of α-SNAP to the CAD domain of Stim1 releases a potential inhibition by the CC1 domain. This can be tested by using recombinant GST–CT Stim1 (already employed in Figure 5) and add increasing amounts of recombinant Orai 1 to obtain a rough estimate of the binding affinity. Addition of increasing amounts of α-SNAP should then increase the affinity of the Stim1–Orai1 reaction or alter the stoichiometry. In a control reaction, α-SNAP could be added to a reaction containing GST–CAD instead of GST–CT Stim1. Assuming that the CAD domain is already active, a-SNAP would not affect the GST–CAD–Orai1 interaction. This experimental set up could provide a mechanistic model of how α-SNAP affects SOCE*.

We agree with the reviewers that, as shown in Figure 5, the binding of α-SNAP to the CAD domain of Stim1 is stronger than interactions with Stim1–CT domains containing the CC1 fragment. However, we are unable to understand how this result, in particular, suggests that α-SNAP relieves a potential inhibition caused by CC1.

We thank the reviewers for suggesting an alternate hypothesis. However, the major problem with testing this hypothesis is the gap in our current understanding of how CC1 mediated inhibition really works and whether the inhibition is relieved in a single molecular step, by Stim1 intrinsic intra-molecular structural changes, or with help from other proteins. Given the long list of potential candidates, this process may depend on a collaborative action of more than one protein.

Furthermore, because Stim1 has been shown to bind both the N- as well as the C-terminal domains of Orai1, only full length Orai1 can ideally be used to test this hypothesis. While we do have GST–CT Stim1, we do not have full-length purified Orai1 (a four trans-membrane protein). Because we do not have experience with multi-pass membrane protein expression, it will likely take us months to purify a reasonable amount of 4 transmembrane protein and we would still be testing a hypothesis that is based on incomplete information about CC1 mediated inhibition.